# Wildfires Improve Forest Growth Resilience to Drought

**Jesús Julio Camarero** [1,*] , **Mercedes Guijarro** [2] , **Rafael Calama** [2] , **Cristina Valeriano** [1] , **Manuel Pizarro** [1]
and **Javier Madrigal** [2,3]

1   Instituto Pirenaico de Ecología (IPE-CSIC), Avda. Montañana 1005, 50059 Zaragoza, Spain;
    cvaleriano@ipe.csic.es (C.V.); m.pizarro@csic.es (M.P.)
2   Instituto de Ciencias Forestales (ICIFOR-INIA), CSIC, Ctra. de La Coruña km 7,5, 28040 Madrid, Spain;
    guijarro@inia.csic.es (M.G.); rcalama@inia.csic.es (R.C.); incendio@inia.csic.es (J.M.)
3   ETSI Montes, Forestal y del Medio Natural, Universidad Politécnica de Madrid (UPM), Ramiro de
    Maeztu s/n, 28040 Madrid, Spain
*   Correspondence: jjcamarero@ipe.csic.es; Tel.: +34-976-369393

**Abstract:** In seasonally dry forests, wildfires can reduce competition for soil water among trees and improve forest resilience to drought. We tested this idea by comparing tree-ring growth patterns of *Pinus pinea* stands subjected to two prescribed burning intensities (H, high; L, low) and compared them with unburned (U) control stands in southwestern Spain. Then, we assessed post-growth resilience to two droughts that occurred before (2005) and after (2012) the prescribed burning (2007). Resilience was quantified as changes in radial growth using resilience indices and as changes in cover and greenness using the NDVI. The NDVI sharply dropped after the fire, and minor drops were also observed after the 2005 and 2012 droughts. We found that post-drought growth and resilience were improved in the H stands, where growth also showed the lowest coherence among individual trees and the lowest correlation with water year precipitation. In contrast, trees from the L site showed the highest correlations with precipitation and the drought index. These findings suggest that tree growth recovered better after drought and responded less to water shortage in the H trees. Therefore, high-intensity fires are linked to reduced drought stress in Mediterranean pine forests.

**Keywords:** dendroecology; growth resilience; Mediterranean forests; *Pinus pinea*; tree rings





## 1. Introduction

Rising temperatures are making some forests more vulnerable to drought damage, increasing dieback incidences and mortality rates, particularly in seasonally dry areas where hotter droughts are frequent [1]. If drought-induced tree mortality increases as tree density and tree-to-tree competition for soil water do, reducing basal area can be an effective management tool to make some forests less vulnerable to drought [2]. This preventive strategy can be achieved through thinning or, in some particular cases, through prescribed fire, which has been shown to reduce drought mortality [3]. However, fire can also increase tree mortality risk by weakening trees and temporarily increasing susceptibility to drought and post-fire insect (bark beetle) attacks [4,5]. Therefore, long-term assessments of post-burning growth responses are required to elucidate these questions.

In the western USA, a century of fire suppression has led to more crowded forests and increased competition for soil water, raising drought stress, particularly in forests that historically experienced high-frequency and low-severity fires [6]. This suggests that fire may reduce drought stress by alleviating competition for soil moisture and nutrients and increasing growth rates or post-drought growth resilience, i.e., the capacity to reach pre-drought growth levels [7]. However, this effect may depend on the season or climate conditions when the burning is performed, on the ecophysiology or xylem phenology of the affected tree species, or even on the on-site conditions [8–15]. For instance, in southeastern Spain, *Pinus halepensis* Mill. forests in dry sites were the most sensitive to the long-term

cumulative impacts of drought and wildfire [14]. Regarding seasonal or climatic factors, burning during fall and after a dry year increased the resilience of *Pinus nigra* ssp. *salzmannii* Dunal (Franco), compared with unburned or pines burned during spring [8]. However, authors did not find a similar impact on burned *Pinus sylvestris* L. trees, which showed a higher resistance (inverse of growth reduction during disturbance) [8].

The value of fire in reducing vulnerability to further disturbances (e.g., drought) is demonstrated by prescribed burning, which is becoming a common practice in Mediterranean, drought-prone forests [16]. It can be a cost-effective adaptation tool, helping forests in warmer and drier future conditions, such as those forecasted for the Mediterranean Basin [17], but the impacts of fire on long-term tree growth are not always tested. In addition, wildfires have multiple effects on forests, such as increasing resource availability by removing understory [12] and releasing soil nutrients [18]; therefore, the adequate fire intensity to reduce tree density or basal area and generate significant post-drought growth resilience is unknown [19]. When considering fire effects on growth recovery after drought, the functional traits of the study species, especially those related to fire tolerance, must also be known and considered. These traits are well characterized in the case of Mediterranean pines, which show different degrees of tolerance and responsiveness to drought and fire [20–22].

Post-drought or post-burning growth resilience or resistance depend on multiple factors in addition to functional traits, including the drought or fire severity experienced by each individual tree (e.g., crown and stem injury, cambium damage, growth reduction, and damage or death of neighbors), characteristics of the drought and wildfire (e.g., intensity, timing, and duration), tree attributes (e.g., species, size, and tree reserves status), and time since the drought or burning [8]. For instance, a very intense burning can damage stem tissues (cambium), causing a long-lasting reduction in hydraulic conductivity through xylem embolism and decreasing the growth recovery capacity after the following drought [11]. This post-fire loss in xylem conductivity has been observed in several conifers [13,23,24], including the Mediterranean stone pine (*Pinus pinea* L.), the focus species of this study. This conifer is considered to have a high tolerance to drought and surface and even crown fires, due to its thick and insulating bark, needles with low sensitivity to extreme heat, high canopy base height, and prolific seeding every 3–4 years [20,21,25]. Its heavy seeds develop during two years, and their abundant reserves facilitate tree recruitment through rapid root elongation in the low-nutrient, acidic, and sandy soils where this species is found [20]. The *P. pinea* scorched crown can recover partially in the spring following the fire [21]. This species achieves reproductive maturity in 10–20 years, shows a short-range of seed dispersal, and does not produce serotinous cones, which limit its regeneration rate in large, burned areas. For these reasons, it is appropriate to assess this species' post-drought growth resilience after fire.

The aims of this study are: (i) to characterize the post-drought growth resilience before and after a wildfire that occurred in 2007, affecting *P. pinea* stands located in the Sierra Morena mountains (Andalusia, southwestern Spain), and (ii) to assess the growth responses to climate and major droughts occurring before (2005) and after (2012) the fire. These aims will be achieved through remote-sensing and field data of fire severity the comparison of climate variables, including a drought index accounting for changes in cumulative water balance, and tree-ring width data obtained through dendrochronological analyses. This will improve the knowledge about fire ecology and the post-disturbance resilience of this species, as well as provide practical information to forest managers on the effects of fires as a way to reduce drought damage. This information can be applied to prescribed burning techniques used to prevent drought impacts. For instance, this study should show if high- or low-severity fires are more or less valuable to providing post-drought growth resilience.

We hypothesize that post-drought growth resilience will be improved in the high-severity fire because of the significant reduction in tree density and competition for soil water.

## 2. Materials and Methods

### 2.1. Study Sites and Wildfire Characteristics

The study sites were located in the Obejo Municipality (Córdoba province, southwestern Spain) near the Guadalmellato reservoir (38.059° N, 4.693° W, 301 m a.s.l.) (Figure 1). Here, a wildfire started on the 27 July 2007 and lasted two days, affecting ca. 1646 ha of the Guadalmellato Site of Community Importance. The site is located within the Sierra Morena *P. pinea* provenance region, where stands of the species develop under Mediterranean climate conditions on acidic soils on substrates such as shales or quartzites [26]. According to climate data from the nearby Guadalmellato reservoir meteorological station, the total annual precipitation was 743 mm, and the mean annual temperature was 17.8 °C [26]. Drought lasted from June to September, with a maximum water shortage in July and August, when mean temperatures reached maximum values (27.4 °C). Mean temperatures reached minimum values in December (10.0 °C). In this area, *P. pinea* coexists with evergreen oaks (*Quercus ilex* L., *Quercus suber* L.), and the understory is dominated by Mediterranean shrubs (*Phillyrea angustifolia* L., *Rhamnus alaternus* L., *Halimium umbellatum* (L.) Spach., *Cistus* spp.).

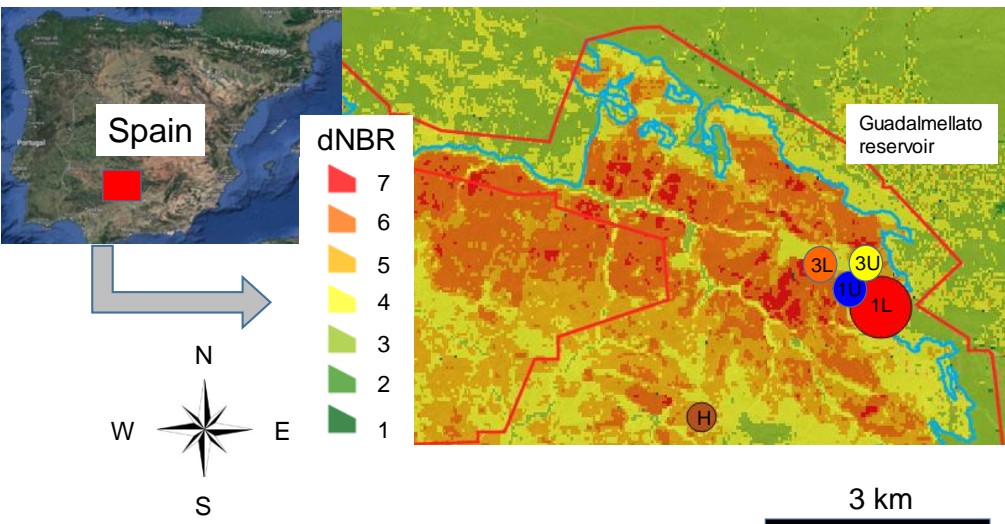

**Figure 1.** Location of the study site in southwestern Spain (red box in the small map) and map showing dNBR values (color scale) near the Guadalmellato reservoir. The locations of the study plots (H, 1L, 3L, 1U and 3U plots) are indicated.

### 2.2. Remote Sensing Information, Climate Data, and Drought Index

The 2007 fire severity was characterized by analyzing Landsat 4–5 satellite images using the differenced Normalized Burn Ratio (dNBR) index [27]. The Normalized Burn Ratio (NBR) calculation is used to highlight burned areas and to estimate the severity of fires. Burned areas have a low NBR value, whereas unburned or healthy vegetation will have a high NBR value.

The Normalized Difference Vegetation Index (NDVI) is a commonly used vegetation index, which measures changes in vegetation cover and greenness, derived from the near-infrared and red channels of remotely sensed imagery [28]. Using the Google Earth Engine, a cloud-based platform for multi-temporal analysis of satellite imagery using the Google computational infrastructure, the NDVI was calculated biweekly for the period 1982–2020 [29,30]. We performed topographic corrections and eliminated cloudy images to obtain the final NDVI values. We also harmonized different Landsat sensors (TM, ETM+ and OLI). Mean NDVI values were calculated for the plots with higher fire severity, according to dNBR (Figure 1), while considering the 50 m circular buffers centered in the selected patches. Lastly, we calculated NDVI differences between consecutive dates (ΔNDVI) to highlight abrupt post-disturbance NDVI changes.

To obtain long-term, homogeneous climate records from the study area, we selected monthly mean maximum and minimum air temperatures and total precipitation data from the 1 km$^2$ gridded E-OBS v. 22.0e database [31]. To evaluate drought intensity, we used the multi-scalar Standardized Precipitation Evapotranspiration Index (SPEI) [32]. Negative and positive SPEI values indicate dry and wet conditions, respectively. The SPEI considers the effect of temperature on a cumulative water balance, which was calculated at 1—(SPE1), 3—(SPEI3), 6—(SPEI6) and 9-month (SPEI9)-long scales. The SPEI was obtained from the 1.1 km$^2$ gridded Spanish SPEI database available at the webpage https://monitordesequia.csic.es/ [33] (last accessed on 15 February 2023). Both climate data and the SPEI were obtained for the period 1975–2021.

## 2.3. Field Sampling Design

Trees surviving the 2007 fire were sampled in spring 2022. Five plots were selected in the field, according to fire severity estimated using the dNBR and information provided by local fire managers (Figure 1): two plots in the low-severity stands (1L, 3L), one plot in the high-severity stands (1H, the unique area that survived high-severity fire in the wildfire), and two plots in the unburned site (1U and 3U plots). The mean post-fire tree density values in plots H, L, and U were 48, 406, and 597 stems ha$^{-1}$, respectively.

To carry out dendrochronological analyses, 15 trees per plot were sampled, and two cores per tree were taken at 1.3 m using Pressler increment borers (n = 150 cores). Trees were characterized by taking dendrometric measures (diameter at 1.3 m, height, crown base height) and proxies of fire damage (scorch height). Crown scorch height was measured in the H (2.09 ± 0.9 m) and L (1.15 ± 0.2 m) plots, since it was easy to visually assess in the field and is widely used as a post-fire mortality predictor in logistic regression models [34]. Trunk scorch height is also widely used as a proxy for damage to the cambium and severity of burning at the tree level [15]. For this study, it was only measured in the H plot, where it was easy to visually asses in the field, which was not the case for the L plots. Bark thickness was also measured with a bark gauge.

## 2.4. Tree-Ring Width Data

We used dendrochronology to quantify changes in radial growth and growth responses to drought, including resilience [35]. Cores were air-dried in the laboratory and glued into wooden supports, and their surface was sanded to enhance the visibility of the tree-ring limits. Then, the cores were scanned at 2400 dpi using a high-resolution scanner (Epson Expression 10.000 XL, Seiko Epson Corp., Matsumoto-shi, Japan). Ring widths were measured with a 0.001 mm resolution using the CooRecorder software [36]. Cross-dating was checked using the CDendro and COFECHA software [36,37], which calculate moving correlations between the individual series and the mean series of each treatment.

## 2.5. Processing Tree-Ring Width Data

To show growth trends, we converted the individual ring-width series into the mean basal area increment (BAI) series, assuming a concentric stem growth and accounting for bark thickness. Differences in tree size and BAI between treatments were assessed using non-parametric Mann–Whitney *U* tests.

To calculate climate–growth relationships, we removed age-, disturbance- or size-related influences on tree-ring width through detrending [35]. Individual tree-ring width series were detrended using a cubic smoothing spline with a 50% frequency response cutoff at 2/3 of the length of the series. Then, autoregressive models were applied to each detrended series to remove the first-order autocorrelation so as to obtain residual or pre-whitened ring-width chronologies. These indexed series were averaged by using a bi-weight robust mean to develop site-level mean series or chronologies for the three study sites. We calculated several statistics to characterize growth variability, namely the first-order autocorrelation (AR1) of the ring-width series, to measure the persistence in growth and the mean sensitivity (MSx), which is the relative change in width between consecutive

rings of indexed, non-pre-whitened data, i.e., standard chronologies [35]. We also obtained the mean inter-series correlation (Rbar), which measures the growth coherence among trees [38]. These variables were calculated over the common and best-replicated period of 1980–2021.

To assess post-drought resilience, we used a standard, i.e., non-pre-whitened, series of ring-width indices and focused on the 2005 and 2012 droughts. We calculated four resilience indices for these two droughts afterwards [39]: Rt, resistance; Rc, recovery; Rs, resilience; and RRs, relative resilience. The four indices were computed as:

$$Rt = RWI_D / RWI_{Pre-D} \tag{1}$$

$$Rc = RWI_{Post-D} / RWI_D \tag{2}$$

$$Rs = RWI_{Post-D} / RWI_{Pre-D} \tag{3}$$

$$RRs = (RWI_{Post-D} - RWI_D) / (RWI_{Pre-D}) \tag{4}$$

where $RWI_D$ corresponds to the standard ring-width index during the drought year, and $RWI_{Pre-D}$ and $RWI_{Post-D}$ correspond to the average indices of the three years before and after the drought, respectively. The resilience components were calculated for every tree in each plot and treatment while also considering the NDVI data. Note that there was a negative relationship between Rt and Rc, as Rs = Rt × Rc [39]. In addition, RRS = Rs − Rt. Differences between treatments in terms of growth resilience were assessed using Mann–Whitney tests.

Climate–growth relationships were based on Pearson correlations between monthly or annual (water year; period between October 1 of year $t-1$ and September 30th of year $t$) climate variables (mean temperature, total precipitation) and residual ring-width indices. Correlations were calculated from the prior to the current September, since climate conditions in the previous year influence subsequent growth [35]. Drought–growth relationships were based on Pearson correlations between the residual chronologies and weekly SPEI data while considering the 1—(SPEI1), 3—(SPEI3), 6—(SPEI6) and 9-month (SPEI9)-long scales.

Detrending, chronology building, and the calculation of dendrochronological statistics were performed using the dplR package [40]. The resilience indices were calculated using the pointRes package [41]. Climate–growth correlations were calculated using the treeclim package [42]. All analyses were carried out using the R statistical software [43].

### 3. Results

*3.1. NDVI Responses to the 2007 Fire and the 2005 and 2012 Droughts*

There was a clear drop in NDVI after the 2007 fire (Figure 2). The annual NDVI passed from a mean value of 0.73 in 2006 to a mean of 0.59 in 2007. The biggest biweekly NDVI drop (ΔNDVI = −0.24) was observed from the second half of July 2007 to the first half of August; this was followed by NDVI drops detected in mid-2005 (−0.19) and 2012 (−0.18) during two severe droughts.

*3.2. Growth Changes after Fire and Drought*

In the trees from the H plot, we found a positive relationship between scorch height and basal area increment for the period 2008–2021 (Figure 3). In fact, trees from the H plot showed higher basal area increment values than trees from the other U and L plots (Table 1). The H-plot trees also presented low AR1 and Rbar values, indicating low year-to-year growth persistence and low coherence in growth among trees (Table 1).

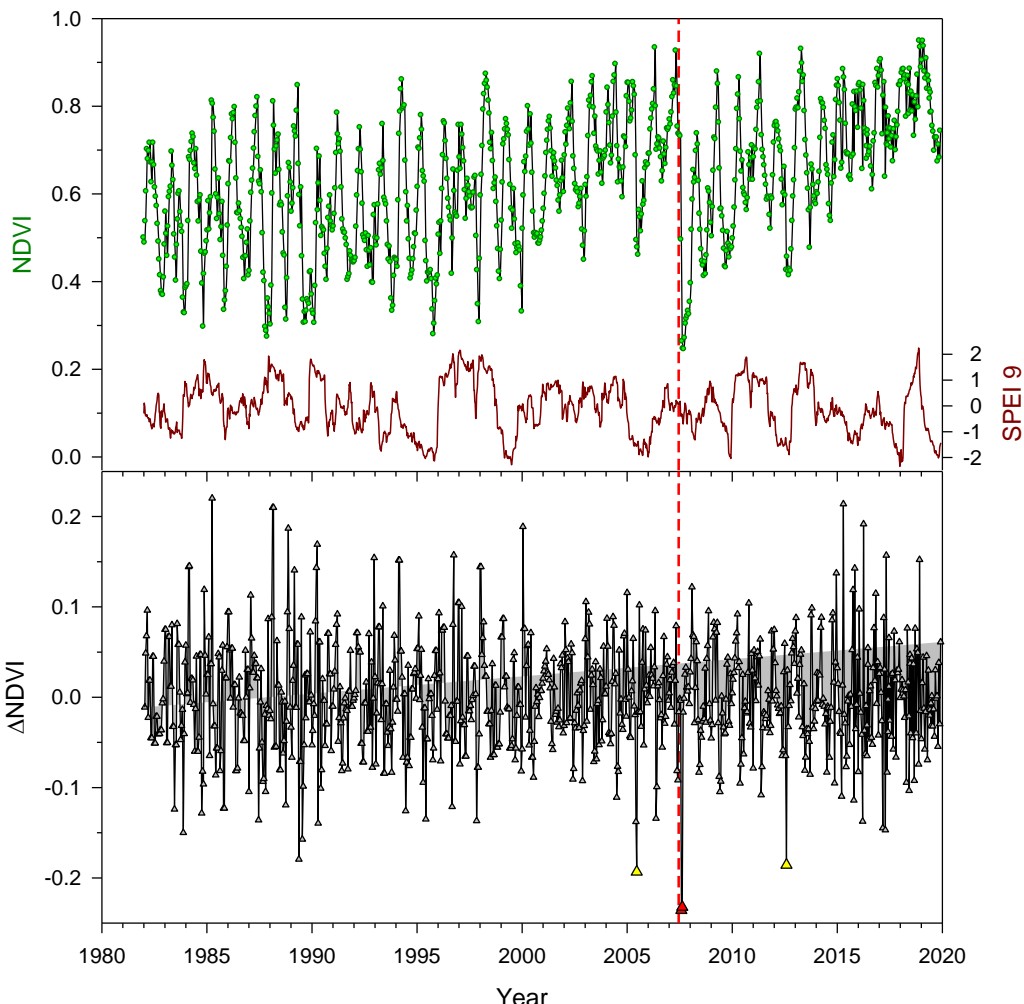

**Figure 2.** Changes in NDVI as related to drought severity (9-month SPEI calculated for September; dark red line, right y-axis). The lowermost plot shows NDVI differences between consecutive dates (yellow triangles correspond to 2005 and 2012 droughts; red triangles indicate the post-fire NDVI drop). The vertical dashed line indicates the 2007 fire.

**Table 1.** Dendrometric and tree-ring width statistics. Values are means ± SD. Dendrometric data correspond to the 15 trees sampled per treatment, and they were taken in 2021. Basal area increment was calculated for the post-fire period (2008–2021). Different letters indicate significant ($p < 0.05$) differences between treatments, according to Mann–Whitney tests.

| Treatment (Code) | Diameter at 1.3 m (cm) | Height (m) | Crown Base Height (m) | Bark Thickness (cm) | Age at 1.3 m (years) | Basal Area Increment (cm²) | AR1 | MSx | Rbar |
|---|---|---|---|---|---|---|---|---|---|
| Unburned (U) | 25.2 ± 4.7 | 14.0 ± 2.1 | 8.5 ± 1.0b | 1.8 ± 0.5 | 49 ± 3 | 3.8 ± 2.1a | 0.80 | 0.42 | 0.71 |
| Low severity (L) | 24.0 ± 3.9 | 11.7 ± 2.2 | 6.9 ± 1.8b | 1.7 ± 0.5 | 50 ± 4 | 3.3 ± 2.5a | 0.78 | 0.46 | 0.62 |
| High severity (H) | 30.0 ± 5.7 | 10.6 ± 1.8 | 3.6 ± 0.8a | 2.5 ± 0.4 | 39 ± 9 | 13.3 ± 4.7b | 0.65 | 0.44 | 0.49 |

We only detected a growth drop (–69.4%) after the 2007 fire in the H plots, where many trees did not form latewood (Figure 4). The growth reductions after the 2012 drought were higher (in absolute terms) in the L-U plots (L, –70.2%; U, –64.1%) than in the H plot (–42.3%). However, during the 2005 drought, prior to the fire, the growth reductions were similar among treatments (H, –47.7%; L, –55.0%; U, –51.4%). Other growth reductions corresponding to dry conditions were also observed in 1988–1989, 1995, and 1999.

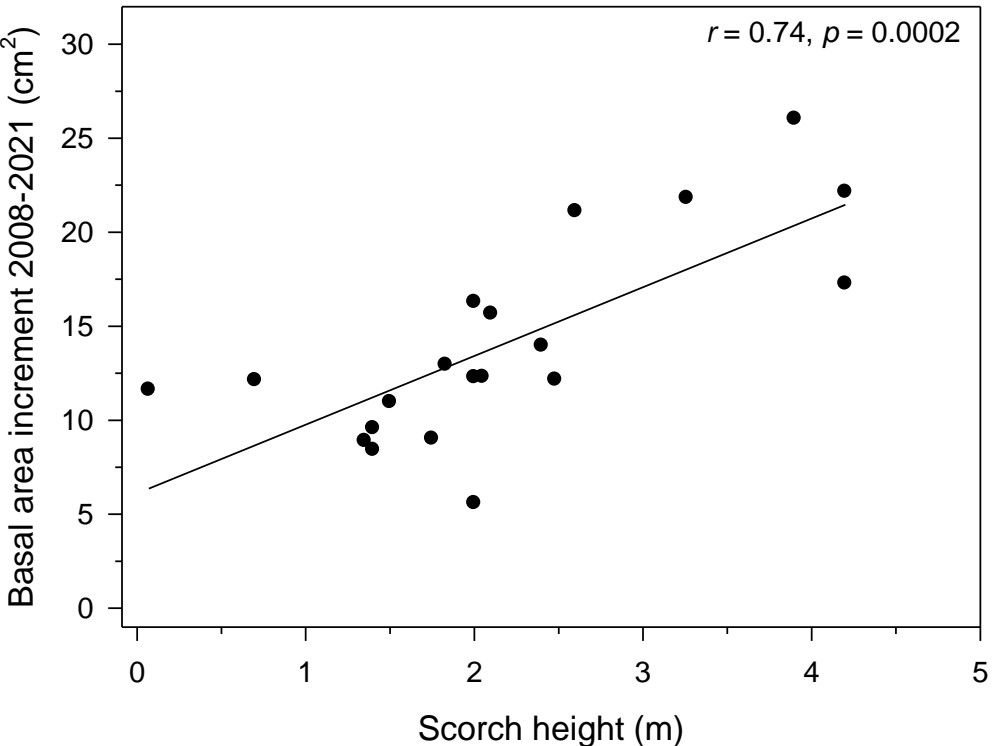

**Figure 3.** Positive relationship observed between scorch height and basal area increment in the post-fire period (2008–2021) of trees subjected to high-severity fires (H plots).

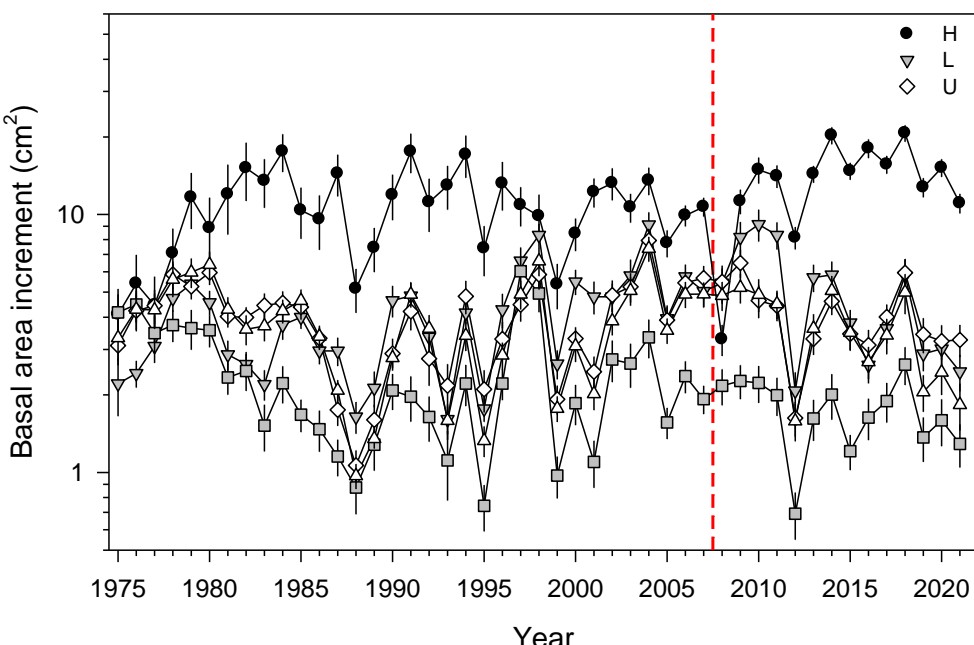

**Figure 4.** Basal area increment series in the three treatments (H, high severity, 1 plot; L, low severity, 2 plots; U, unburned, 2 plots). The vertical dashed line shows the 2007 fire. Note the growth drops in the 2005 and 2012 droughts. Values are means ± SE.

### 3.3. Growth Responses to Climate Variables and the Drought Index

Growth was constrained by cold conditions in February (only in the L plot), by warm conditions in the previous September (only in the L plot), and in the current May (plots L and U) (Figure 5). Growth of trees from plot L was enhanced by high precipitation in the hydrological year, in the prior autumn and winter, and in May. Trees from the U and H

plots also grew more with higher precipitation during the water year and in February in the case of U. However, the strongest correlation between growth indices and precipitation of the water year ($r = 0.68$, $p < 0.001$) was found in the L plot (Figure 6).

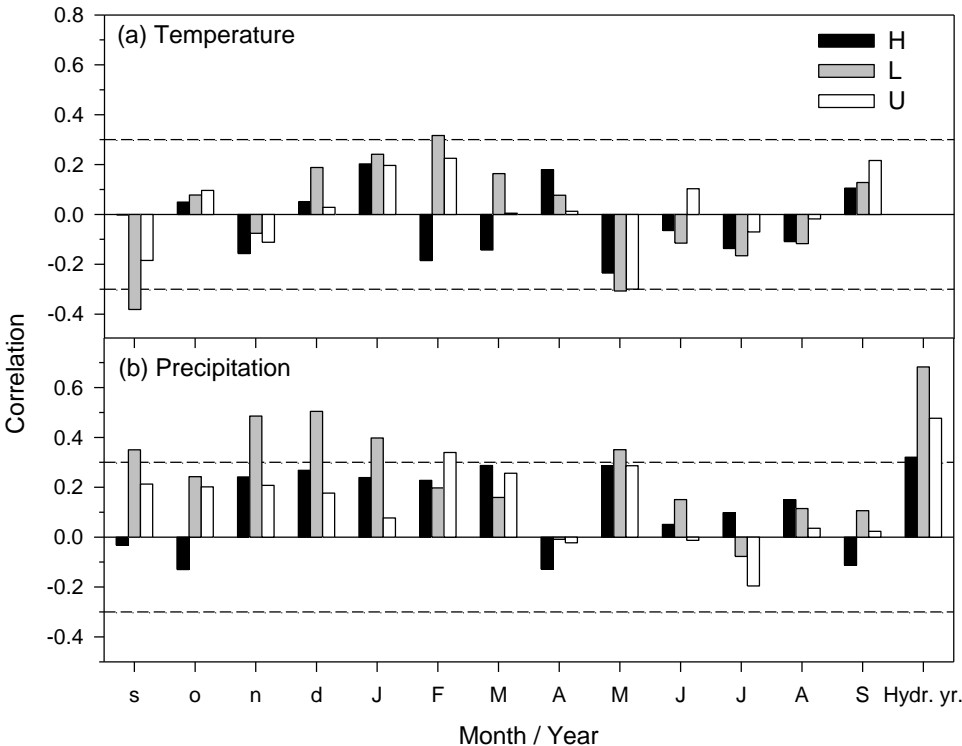

**Figure 5.** Correlations calculated by relating monthly or annual climate variables (mean temperature, total precipitation) and series of ring-width indices of the three treatments (H, L, U). Months abbreviated by lower- and uppercase letters correspond to the prior and current years, respectively. The dashed horizontal lines show the 0.05 significance levels.

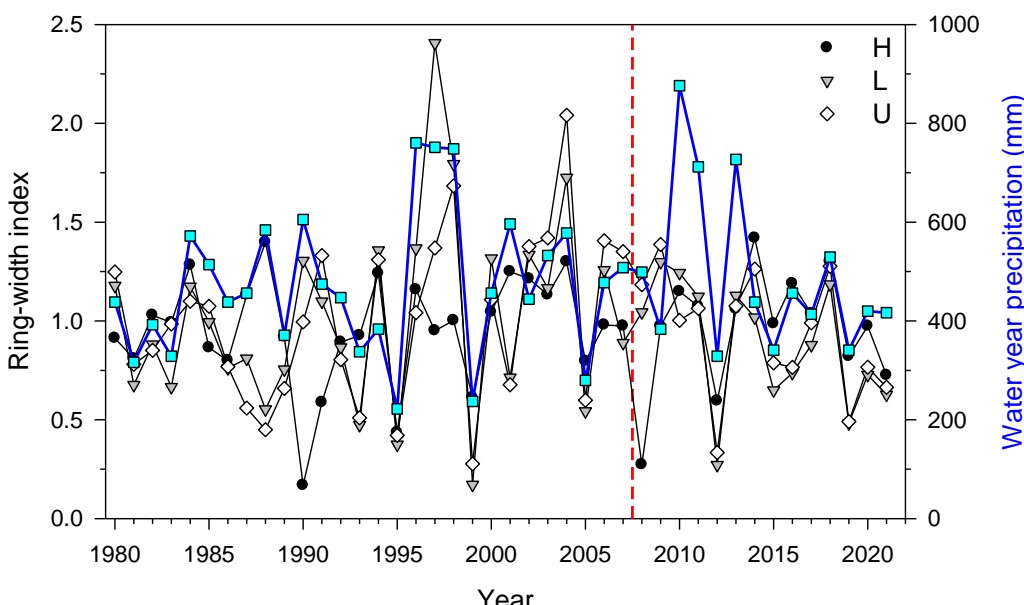

**Figure 6.** Mean series of ring-width indices in the three treatments (H, L, U) and precipitation of the hydrological year (right y-axis, blue symbols, and line). The vertical dashed line shows the 2007 fire.

The correlations between growth indices and the SPEI were higher in the L than in the U and H treatments, in this order (Figure 7). In the series from H, no significant correlation was found with SPEI. In the case of L, correlations peaked in early June in the case of the 6-month SPEI (SPEI6), whilst in the case of U, correlations peaked in late February (SPEI1) or mid-April (SPEI9).

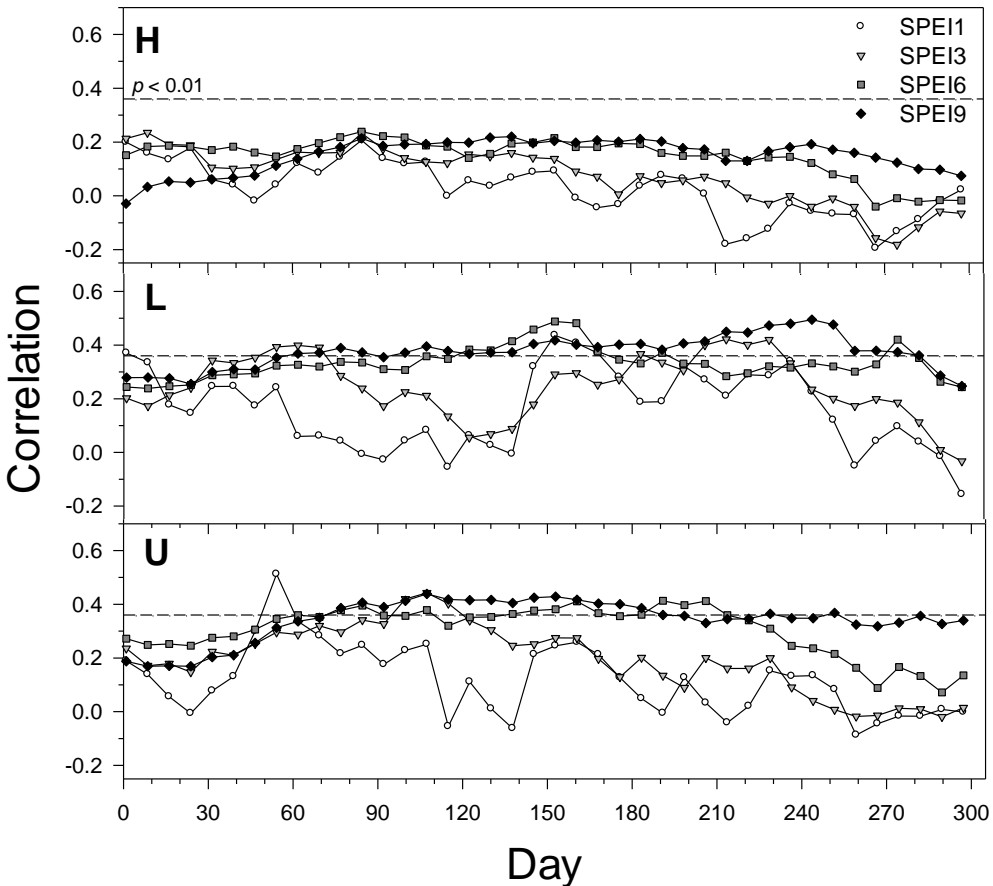

**Figure 7.** Correlations obtained by relating the series of ring-width indices of the study plots (H, L and U) and weekly SPEI data while considering 1—(SPEI1), 3—(SPEI3), 6—(SPEI6), and 9-month (SPEI9)-long scales. The horizontal dashed lines indicate the 0.01 significance levels. Day 0 is the 1st of January.

### 3.4. Resilience Indices

The post-fire BAI values were positively correlated with the Rt indices of the 2005 ($r = 0.33$, $p = 0.01$) and 2012 droughts ($r = 0.69$, $p < 0.001$) but negatively correlated with the corresponding Rc indices (2005, $r = -0.39$, $p = 0.02$; 2012, $r = -0.26$, $p = 0.05$). The BAI values were not correlated with the 2005 resilience indices (Rs, RRs), but they were correlated with the 2012 indices (Rs, $r = 0.47$, $p < 0.001$; RRs, $r = 0.31$, $p = 0.02$).

No significant differences in the resilience indices were found between plots in the 2005 drought, but Rt, Rs, and RRs were significantly ($p < 0.05$) higher in the H plot than in the other plots during the 2012 drought (Figure 8). Resilience indices of NDVI showed similar values between the two droughts, but Rc, Rs, and RRs showed higher values in the 2012 drought than in the 2005 drought.

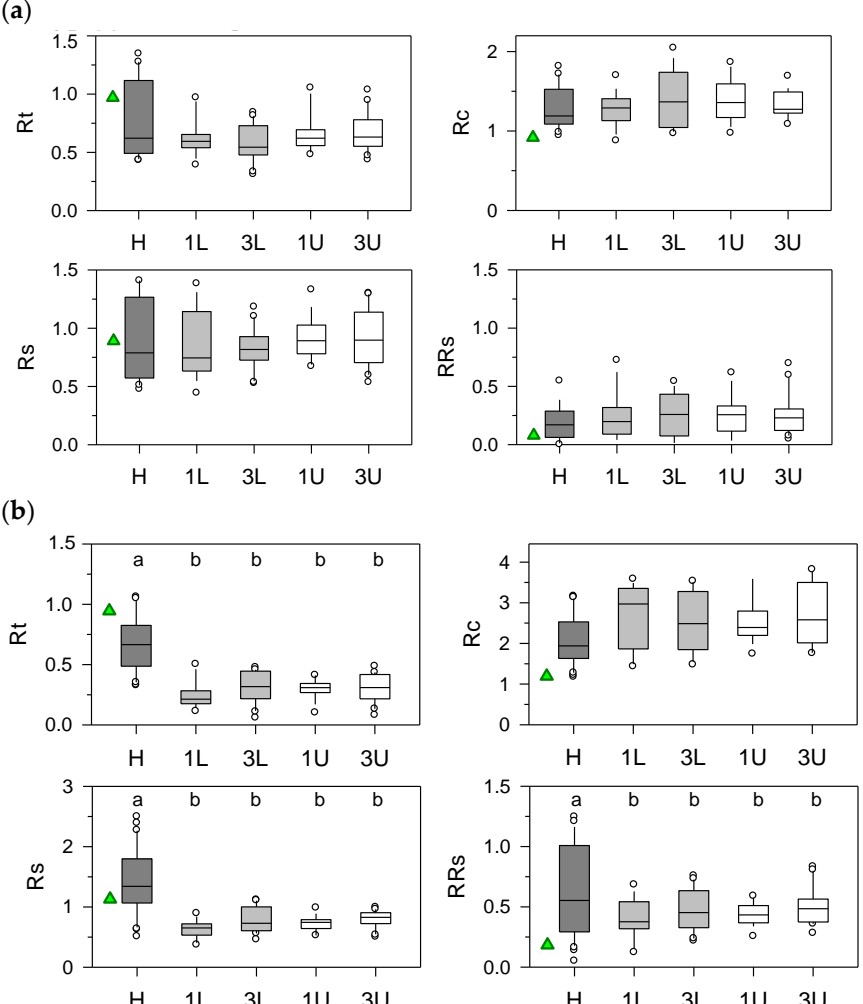

**Figure 8.** Resilience indices (Rt, resistance; Rs, resilience; Rc, recovery; RRs, relative resilience) calculated from annual ring-width indices (box plots) or NDVI data (green triangles) while considering the (**a**) 2005 and (**b**) 2012 droughts. The box plots correspond to the two fire severities (H, high; L, low) and the unburned (U) plots. In the L and U treatments, data for plots 1 and 3 are presented. Different letters indicate significant (*p* < 0.05) differences between treatments according to Mann–Whitney tests.

## 4. Discussion

The hypothesis was supported by our data, since *P. pinea* showed a high growth resilience to the 2012 drought in the plots subjected to high fire severity (H), with higher values of growth resistance (Rt), resilience (Rs), and relative resilience (RRs) than the low-severity fire (L) and unburned (U) plots. This means that: (i) growth was less negatively impacted (higher resistance) by the 2012 drought in the H plot than in the L and U plots, and (ii) growth recovered better (higher resilience and relative resilience) in the H plot. Such enhanced post-drought growth concurred with the higher growth responsiveness to the water year precipitation and the SPEI drought index in the L more than in the H plots. In plots burned by low-severity fire, competition for soil water was stronger, and tree growth responded more to precipitation and soil moisture than in plots subjected to high-severity fires.

Interestingly, in *P. pinea* subjected to prescribed burning, burned trees did not suffer mechanical damage nor a reduction in radial growth or changes in water-use efficiency, which suggest a high resistance of this species to surface fires, due to its thick, insulating bark [13,25,44]. However, lower values of $\delta^{18}O$ were measured in the tree-ring wood of burned trees, which was interpreted as reduced competition for soil water and nutrients,

which increased both photosynthetic activity and stomatal conductance. Since the presented bark thickness measurements were comparable to other studies on *P. pinea* [13] and no cambium necrosis or abnormal tracheids were observed in the cores, it was plausible that the growth reduction caused by the fire was due to crown scorch and damage, as inferred in other species, such as *P. nigra* [15]. Indeed, in southern France, crown scorch volume was the variable that best explained the growth decline and mortality of *P. pinea* after wildfires [44]. The higher bark thickness observed in the H plot, compared with the L and U plots, suggested that fire could improve the resilience to new events by the inversion of the tree in bark tissues [20] or that individual trees with higher bark thickness before the fire increased their survival rate [25]

In Californian conifers, increasing crown scorch was also associated with a greater risk of post-fire mortality, but trees with substantial crown damage were more vulnerable to delayed mortality if also exposed to drought [45]. These findings suggested an interaction between fire damage and vulnerability to drought stress, which acts on the tree and stand scales and should be investigated while considering the long (decadal) temporal scales. This is because the responses of individual trees to fire and drought are complex and partially dependent on local competitive environments and on cumulative impacts of past fire and drought events [46]. A shortcoming of our analyses is the lack of tree-level competition measures before and after the fire and the existence of only one H plot (the unique area burned that survived the high-severity fire). In addition, future studies should consider removing or preserving the understory to account for its role as a source of water and nutrients. However, we are confident about the robustness of our results, given the high number of analyzed trees and the coherent year-to-year growth variability, likely driven by precipitation variability, revealed by tree-ring data.

Our results agree with previous field observations in national parks of the Sierra Nevada (California, USA), where burned sites had lower stem density and lower proportions of recently dead trees than unburned sites after drought [47]. That study suggested that prescribed fires and/or thinning (mechanical) treatments lessened competition for water (with other trees or with understory vegetation) so that surviving trees might be more likely to resist and recover after drought. Our tree-ring data were in line with this idea, confirming that growth recovery was improved in the plot subjected to a high-severity fire. This is especially relevant in species such as *P. pinea,* which form a xylem vulnerable to drought-induced embolism and whose radial growth is greatly constrained by low precipitation [48–50].

## 5. Conclusions

High-severity wildfire increased the growth resilience to drought in *Pinus pinea*. This improved growth recovery after water shortage was likely mediated by a reduction in tree density and tree-to-tree competition for soil water. Our findings suggest that wildfires improve forest growth resilience to drought in seasonally dry regions, such as the Mediterranean Basin. These results can be tested in other sites while considering additional species with contrasting resistance to fire and resilience to drought and performing different prescribed burning treatments to reduce tree density in stands vulnerable to drought damage. The roles of fires of different severity as sources of resilience to drought should be further tested and refined.

**Author Contributions:** Conceptualization, J.J.C. and J.M.; methodology, M.G., R.C., C.V. and M.P.; software, C.V. and M.P.; validation, M.G., R.C. and J.M.; formal analysis, J.J.C.; investigation, J.M.; resources, J.M.; data curation, M.G., R.C., C.V. and M.P.; writing—original draft preparation, J.J.C.; writing—review and editing, J.M., M.G., R.C., C.V. and M.P.; visualization, J.M.; supervision, J.M.; project administration, J.M.; funding acquisition, M.G., J.M. and J.J.C. All authors have read and agreed to the published version of the manuscript.

**Funding:** This research was funded by the Spanish Ministry of Science and Innovation projects (grant number TED2021-129770B-C21 project, grant number PID2020-116494RR-C41 project MCIN/AEI/

10.13039/501100011033/, VIS4FIRE project grant number RTA2017-00042-C05-01) and Interreg-POCTEP CILIFO (Iberian Centre for Research and Forest Firefighting; grant number 0753-CILIFO-5-E).

**Data Availability Statement:** The data presented in this study are available upon reasonable request to the corresponding author.

**Acknowledgments:** We are grateful to the Fire Service of the regional government of the Andalusia Autonomous region for providing field assistance and the Forest Fire Laboratory of Universidad de Córdoba for providing stand data. We thank Sara de Paula and Sandra Nicolás (ICIFOR-INIA, CSIC) for the field sampling and laboratory work.

**Conflicts of Interest:** The authors declare no conflict of interest.

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
