# Peer review of "Wildfires Improve Forest Growth Resilience to Drought"

_fire, doi:10.3390/fire6040161_

Round 1
Reviewer 1 Report
1. Figure1 should be redraw, and add compass in it.
Author Response
Comments and Suggestions for Authors
- Figure1 should be redraw, and add compass in it.
- Thanks for your comment. We have redrawn Figure 1 and added the compass.
Reviewer 2 Report
The authors show that for their study areas, the previously burned areas experienced less mortality than the unburned areas. I'll conditionally accept that. My problem is that I expect that there was earlier mortality associated with that prescribed fire. So given an observation window of 50 or 100 years, counting mortality prior as well as following the burn treatment(s), does fire actually promote greater survival? I understand that the authors may not have information to answer that. That's fine, but should not this be explained as part of the current uncertainty?
A few specific line comments as well:
|
Lines |
Notes |
|
34-35 |
The cited reference (#3) by van Mantgem and others does show reduced drought mortality for sites which had been previously treated with fire, but I don’t think that those authors attributed that to conferred “resistance to water shortage”. Need to check that that inference was expressed in that earlier paper. |
|
41-42 |
By “releasing from competition” do the authors mean from earlier periods of mortality? |
|
66 |
“Resistance” to what? |
|
362-363 |
What or which is being referred to as “which was resistant to surface fire”? As written, could be the tree species or “the growth resilience to drought”. |
Author Response
Comments and Suggestions for Authors
The authors show that for their study areas, the previously burned areas experienced less mortality than the unburned areas. I'll conditionally accept that. My problem is that I expect that there was earlier mortality associated with that prescribed fire. So given an observation window of 50 or 100 years, counting mortality prior as well as following the burn treatment(s), does fire actually promote greater survival? I understand that the authors may not have information to answer that. That's fine, but should not this be explained as part of the current uncertainty?
- Sorry, but we did not study post-fire changes in tree mortality but on growth resilience to drought. We agree on the reviewer’s comment, but answering that issue is out of the scope of the study since we focused on changes in radial growth and not on mortality rate.
A few specific line comments as well:
|
Lines |
Notes |
|
34-35 |
The cited reference (#3) by van Mantgem and others does show reduced drought mortality for sites which had been previously treated with fire, but I don’t think that those authors attributed that to conferred “resistance to water shortage”. Need to check that that inference was expressed in that earlier paper. |
|
41-42 |
By “releasing from competition” do the authors mean from earlier periods of mortality? |
|
66 |
“Resistance” to what? |
|
362-363 |
What or which is being referred to as “which was resistant to surface fire”? As written, could be the tree species or “the growth resilience to drought”. |
- We rephrased the sentence related to reference #3.
- We mean that fire may reduce tree-to-tree competition for soil water and nutrients. We rephrased the sentence.
- We rephrased the sentence to clarify its meaning.
Reviewer 3 Report
View letter
1) Title: Wildfires maybe better.
2) Abstract, the conclusion, “Therefore, high-intensity fires may alleviate drought stress in Mediterranean pine forests.”. This view is a little beyond the recognition of forest system conservation.
3) Keyword: I am confused about the word “prescribed burning”.
4) Line 106, how long did the wildfire last?
5) Line 153, How to define high and low severity stands?
6) Line 157, how far apart are the trees in different treatments?
7) In Fig. 5, why the temperature correlations were different for three-degree treatments? Shouldn't the temperature be the same? They're all in the same area. Careful about the short for month in x-coordinate.
8) In Fig. 7, the day starts from which day?
9) 3.4 Resilience indices, although the difference is significant, the correlation coefficient is not high, and may still not be close with each other.
10) Line 325, it would be better to have soil moisture data here.
11) Line 348, whether the trees themselves are more different in different regions?
12) Conclusion, does the natural death of weak trees help other trees resist drought? The fire's effect should be similar, changing forest density. Does it follow from this that forest fires need not be controlled or even increased is beneficial for forest system?
13) Check the references form.
Author Response
Comments and Suggestions for Authors
1) Title: Wildfires maybe better.
- Ok, we rephrased the title.
2) Abstract, the conclusion, “Therefore, high-intensity fires may alleviate drought stress in Mediterranean pine forests.”. This view is a little beyond the recognition of forest system conservation.
- Thanks for your comment. We rephrased the sentence.
3) Keyword: I am confused about the word “prescribed burning”.
- We removed it.
4) Line 106, how long did the wildfire last?
- We added this information.
5) Line 153, How to define high and low severity stands?
- They we defined based on remote sensing (dNBR; lines 118-119) and site (reports by forest technicians) information.
6) Line 157, how far apart are the trees in different treatments?
- Sorry, but we did not measure tree-to-tree distances but mean stand density after the fire.
7) In Fig. 5, why the temperature correlations were different for three-degree treatments? Shouldn't the temperature be the same? They're all in the same area. Careful about the short for month in x-coordinate.
- Post-fire changes in tree density can also affect growth correlations with temperature even if trees are located in the same area. For instance, competition for water in L plots can explain the strongest negative correlation with prior September temperatures.
8) In Fig. 7, the day starts from which day?
- Day 0 is the 1st We indicated in the revised legend.
9) 3.4 Resilience indices, although the difference is significant, the correlation coefficient is not high, and may still not be close with each other.
- Displayed correlations were significant at the 0.05 level.
10) Line 325, it would be better to have soil moisture data here.
- We agree. Regrettably, these data are not available.
11) Line 348, whether the trees themselves are more different in different regions?
- We can’t answer that question. We showed the high common growth variability among trees of different treatments and also in the study site. We do not know if the same growth coherence occurs among other regions, albeit this is possible given that tree growth is mainly controlled by water availability in Mediterranean forests.
12) Conclusion, does the natural death of weak trees help other trees resist drought? The fire's effect should be similar, changing forest density. Does it follow from this that forest fires need not be controlled or even increased is beneficial for forest system?
- We rephrased the conclusions by answering your questions. We think that tree mortality helps other surviving trees to resist drought, but this was not the object of our study. We do not think that tree death processes have a similar effect as wildfire since they have very different spatial patterns. Tree death may be patchy, whereas wildfires mainly affect some more exposed stands. Our conclusions indicate that high-severity fires may alleviate drought stress; so they could be “imitated” in some areas to reduce tree density through prescribed burning.
13) Check the references form.
- Done, thank you.
Reviewer 4 Report
Very interesting and useful work that investigate "seems to be obvious but poorly explored statement" and shows positive role of wildfires.
Author Response
Comments and Suggestions for Authors
Very interesting and useful work that investigate "seems to be obvious but poorly explored statement" and shows positive role of wildfires.
- We thank you for your positive comments on our study.